# Synapsin-Promoted Caveolin-1 Overexpression Maintains Mitochondrial Morphology and Function in PSAPP Alzheimer’s Disease Mice

**DOI:** 10.3390/cells10092487

**Published:** 2021-09-20

**Authors:** Shanshan Wang, Taiga Ichinomiya, Yuki Terada, Dongsheng Wang, Hemal H. Patel, Brian P. Head

**Affiliations:** 1Veterans Affairs San Diego Healthcare System, 3350 La Jolla Village Drive, San Diego, CA 92161, USA; shw049@ucsd.edu (S.W.); taiga@nagasaki-u.ac.jp (T.I.); y-1221@naramed-u.ac.jp (Y.T.); dow019@ucsd.edu (D.W.); 2Department of Anesthesia, University of California San Diego, San Diego, CA 92093, USA; 3Department of Anesthesiology and Intensive Care Medicine, Graduate School of Biomedical Sciences, Nagasaki University, Nagasaki 8528501, Japan; 4Department of Anesthesiology, Nara Medical University, Kashihara 6348521, Japan

**Keywords:** caveolin, gene therapy, mitochondria, oxidative stress, transgenic, Alzheimer’s disease

## Abstract

Mitochondrial dysfunction plays a pivotal role in the Alzheimer’s Disease (AD) pathology. Disrupted mitochondrial dynamics (i.e., fusion/fission balance), which are essential for normal mitochondria structure and function, are documented in AD. Caveolin-1 (Cav-1), a membrane/lipid raft (MLR) scaffolding protein regulates metabolic pathways in several different cell types such as hepatocytes and cancer cells. Previously, we have shown decreased expression of Cav-1 in the hippocampus of 9-month (m) old PSAPP mice, while hippocampal overexpression of neuron-targeted Cav-1 using the synapsin promoter (i.e., *SynCav1*) preserved cognitive function, neuronal morphology, and synaptic ultrastructure in 9 and 12 m PSAPP mice. Considering the central role of energy production in maintaining normal neuronal and synaptic function and survival, the present study reveals that PSAPP mice exhibit disrupted mitochondrial distribution, morphometry, and respiration. In contrast, *SynCav1* mitigates mitochondrial damage and loss and enhances mitochondrial respiration. Furthermore, by examining mitochondrial dynamics, we found that PSAPP mice showed a significant increase in the phosphorylation of mitochondrial dynamin-related GTPase protein (DRP1), resulting in excessive mitochondria fragmentation and dysfunction. In contrast, hippocampal delivery of *SynCav1* significantly decreased p-DRP1 and augmented the level of the mitochondrial fusion protein, mitofusin1 (Mfn1) in PSAPP mice, a molecular event, which may mechanistically explain for the preserved balance of mitochondria fission/fusion and metabolic resilience in 12 m PSAPP-*SynCav1* mice. Our data demonstrate the critical role for Cav-1 in maintaining normal mitochondrial morphology and function through affecting mitochondrial dynamics and explain a molecular and cellular mechanism underlying the previously reported neuroprotective and cognitive preservation induced by *SynCav1* in PSAPP mouse model of AD.

## 1. Introduction

Alzheimer’s Disease (AD) is characterized by abundant amyloid-β (Aβ) plaques, neurofibrillary tau tangles, mitochondria dysfunction, disrupted synaptic signaling, and loss of synapses, all of which contribute to neuronal degeneration and loss. While the accumulation of damaged mitochondria may result from Aβ and phosphorylated tau pathology, recent preclinical and clinical studies suggest that mitochondria dysfunction alone can lead to Aβ oligomeric or fibrillar formation and phosphorylation tau accumulation [1], suggesting that mitochondrial dysfunction may occur earlier and be a more primary cause of AD. Given the high-energy demand of neurons coupled with their complex cellular structure, proper mitochondrial dynamics and function are especially important for neurons to allow transport and function of mitochondria at critical sites such as the synapse [2].

Caveolin-1 (Cav-1) is a membrane/lipid raft (MLR), cholesterol-binding, and scaffolding protein that organizes synaptic signaling components, including neurotrophin and neurotransmitter receptors in synaptic membranes [3,4]. However, recent evidence shows that Cav-1 is also involved in metabolic alterations in several different cell types, such as hepatocytes and cancer cells [5,6,7,8]. Other isoforms of caveolin (e.g., Cav-3) have been shown to localize to mitochondria and modulate mitochondrial structure and function in stress adaptation [9,10]. Similarly, MLRs and Cav-1 have been found to localize in mitochondria in different cell types, including epithelial cells revealed by electron microscopy (EM) [11]. Moreover, in mitotic cells, Cav-1 is upregulated in response to oxidative stress and may serve as a protective mechanism [12,13,14,15,16]. In cancer cells and fibroblasts, oxidative stress stimulates Cav-1 upregulation. This upregulated Cav-1 activates several signaling pathways essential to mitochondrial integrity and function, which in turn mitigates bioenergetic dysfunction [17,18,19]. As Cav1 is ubiquitously expressed and is regulated by oxidative stress, it remains unclear how Cav-1 expression is impacted in the nervous system.

In neurodegenerative diseases, loss of neuroplasticity is coupled to the downregulation of Cav-1 expression. For example, downregulation of Cav-1 transcript was observed in degenerated neurons in CTE patients [20]. Furthermore, we have also observed decreased Cav-1 in the spinal cord of hSOD1G93A mice (i.e., amyotrophic lateral sclerosis or ALS) and in the hippocampus of PSAPP mice (i.e., AD), while neuron-targeted Cav-1 expression using the synapsin promoter (*SynCav1*) significantly extended survival and mitochondria count and morphology in the ALS spinal cord [21] and maintained cognitive function, neuronal morphology, and synaptic ultrastructure in PSAPP mice [22]. Mounting evidence suggests an important role played by mitochondria dysfunction in AD. Mitochondria not only provide ATP for axonal transport and neurite growth, synaptic-located mitochondria also provide energy that maintains synaptic function and mobilization of pre-synaptic vesicles, cellular events essential for neurotransmission. While the neuronal membrane phospholipids are composed of mostly polyunsaturated fatty acids, neuron system is particularly vulnerable to oxidative stress. Thus, the dysfunction of mitochondria in neurons can cause synaptic dysfunction and cognitive decline, pathological changes observed in AD [23]. Considering the pivotal role of oxidative stress and mitochondrial dysfunction to AD neuropathology [24,25,26,27], the significant decreased Cav-1 expression, specifically in neural systems, may exacerbate the ability of neurons to adapt to oxidative stress resulting in mitochondrial dysfunction and subsequent neurodegeneration. The possible mechanism of *SynCav1*-induced neuroprotection in the setting of severe oxidative stress may in part occur through the preservation of mitochondrial morphology. The focus of the current study was to investigate the direct regulatory role of Cav-1 on mitochondrial function and dynamics in PSAPP mice.

## 2. Materials and Methods

### 2.1. Animals

AD-transgenic (Tg) (APPSwePS1d9 a.k.a. PSAPP) and C57BL/6 mice were purchased from Jackson Laboratory (Bar Harbor, ME, USA) and bred inhouse. Transgene negative (WT) littermates were used as controls. Mice were reared (3–5/cage) with free access to food and water. Brain tissue was processed for biochemistry, electron microscopy, and oroboro assay at either 9 m (early symptomatic) and 12 m (severely symptomatic).

### 2.2. Stereotactic Injection

Mice (3 m) were mounted onto a stereotaxic frame under anesthesia (2% isoflurane). Bilateral burr holes were made by a 22-gauge needle. Hippocampal injections using 33-gauge, 10 μL gas tight syringe (Hamilton, Reno, NV, USA) were controlled by Injectomate (RWD, Shenzhen, China). A total of 1.5 μL of adeno-associated virus serotype 9 (AAV9) (viral titer: 109 genome copies (g.c.)/μL) containing synapsin-red fluorescent protein (*SynRFP*) or *SynCav1* was injected bilaterally into the hippocampal region over 180 s at 3 locations (1st site: AP: 1.82 mm, Lat: 1.15 mm, DV: 1.7 mm: 2nd site: AP: 2.30 mm, Lat: 2.25 mm, DV: 1.75 mm; 3rd site: AP: 2.80, Lat: 2.5 mm, DV: 2.00 mm) with 1 min indwelling time, as previously described [20,22]. A representative image of RFP expression was shown in Appendix A.

### 2.3. Immunoblot Analysis

Hippocampi were homogenized at 4 °C in 500 mM sodium carbonate (pH 11.0) (containing protease and phosphatase inhibitors) followed by sonication (10 s × 3 times with 10 s interval). Lysates were then immunoblotted using primary antibodies against Cav-1 (Cell Signaling #3238; 1:1000), p-DRP (Ser616) (Thermo Fisher; #PA5-64821, 1:500), DRP1 (cell signaling, #8570, 1:1000), mitofusion-1 (cell signaling, #14739, 1:500), mitofusion-2 (cell signaling, #9482, 1:1000), Tom20 (cell signaling, #42406, 1:1000), Oxphos (abcam, #ab110413, 1:2000) and GAPDH (Cell Signaling #2118s; 1:2000) overnight at 4 °C followed by incubation with IR-dye labeled secondary antibody for 1 h at room temperature. Membranes were then visualized by LiCor Odyssey followed by densitometric analysis.

### 2.4. Electron Microscopy

Previously, we observed significant hippocampal synaptic loss and learning deficits in 9 m PSAPP and stunted dendritic arborization and progressive memory deficits at 12 m [22]. Thus, for the current study we chose 9 m and 12 m PSAPP mice for EM analysis. EM was performed as previously described [28]. Brains were fixed with 2% glutaraldehyde, treated with 1% OsO4, and were then bloc stained with uranyl acetate. For flat embedding of the sections, thin flexible molds were employed to lay the sections as level as possible in LX112 embedding media and then overlayed with plastic coverslips. The flat embedded blocks were visually trimmed to CA1 regions encompassing distal apical dendrites in the “stratum radiatum” of the hippocampus and thin sectioned to 70 nm. Grids were viewed unstained using FEI Tecnai EM scope. Mitochondria located within CA1 were used for analyses. Mitochondria were defined by the presence of cristae and a defined outer membrane [21]. Excitatory synapses were identified by the presence of a prominent, asymmetric post-synaptic density (PSD) [22]. Mitophagy event was defined by the identical autophagosome structure [29,30], the early stage of mitophagy (autophagosome with engulfed mitochondria) can be easily detected through the identification of unique mitochondrial structures such as the cristae (Appendix A). The late stage of mitophagy may be reflected by the single-membrane autolysosomes with residual mitochondria on the basis of their similar electron density with mitochondria (Appendix A). Damaged mitochondria were defined by disorganized cristae, matrix swelling and vacuoles, and abnormal distension between inner and outer mitochondrial membranes. Approximately 5 micrographs at 11,000 magnification (8–12 filed per animal) per animal (4–6 animals per group) were used to perform the analyses. Approximately 150 total mitochondria per group were used in the analyses for area, perimeter and mitochondria count for mitochondria. Micrographs at 9300 magnification (higher magnification contained less mitochondria per micrograph) were used to analyze for the mitochondrial area and count, mitochondrial fusion/fission, and mitophagy event. Mitochondrial size and count were measured using Adobe Photoshop (San Jose, CA, USA) as previously described [21].

### 2.5. High-Resolution Mitochondrial Respirometry

Hippocampal mitochondrial function was measured using an Oroboros O2k-Respirometer [31]. Briefly, oxygen polarography was performed at 37 °C, and oxygen flux per tissue mass (pmol O_2_/second/mg) was recorded using Datlab software (OROBORO INSTRUMENTS, Innsbruck, Austria). A total of 5 mL oxygen was injected into each chamber for oxygenation at the beginning of each experiment until oxygen concentration reached 380 μM. Hippocampal tissue was homogenized in cold Miro5 media on ice containing: 0.5 mM EGTA, 3 mM MgCl_2_, 60 mM K-lactobionate, 20 mM Taurine, 10 mM KH_2_PO_4_, 20 mM HEPES. 110 mM sucrose and 1.0 g/L fatty acid free BSA; pH 7.1. One milligram of lysate was loaded into each of the 2 mL respirometers chambers containing MiRO_5_ and kept at 37 °C during the measurement. As shown in Figure 1, after 10 min of equilibration, OXPHOS—capacity of complex I (CI OX) linked activity was measured by adding CI-linked substrate pyruvate (5 mM), glutamate (10 mM), malate (0.5 mM), and ADP (2.5 mM); maximum oxidative phosphorylation capacity of CI and II (CI + II OX) linked activity was measured by addition of succinate (10 mM), maximum uncoupled capacity (CI + II mUC) were determined following stepwise titration of 2-[2-[4-(trifluoromethoxy)phenyl]hydrazinylidene]-propanedinitrile (FCCP). Additionally, CII coupled respiration capacity (CII) was determined in another O2k-Respirometer chamber by addition of CI inhibitor, rotenone (0.5 μM), CII substrate succinate (10 mM), and ADP (2.5 mM). Similarly, maximum uncoupled capacity was determined following stepwise titration of 2-[2-[4-(trifluoromethoxy)phenyl]hydrazinylidene]-propanedinitrile (FCCP) (Figure 1B). Oxygen concentration in the chamber was kept at more than 150 μM till the end to avoid the limitation of respiration. Duplicate data of each sample were averaged to obtain the final readout. Oxygen flux was expressed as mass-specific respiration (per mg), or as Flux Control Respiration (FCR) by internal normalization for maximum ETS capacity (CI&II ETS, mUC value) [32]. Mass-specific oxygen consumption rate and FCR was obtained using DatLab 7 software.

### 2.6. Statistical Analyses

Data were checked for normal distribution and analyzed by Student *t*-tests, one-way analysis of variance (ANOVA), or 2-way ANOVA followed by Fisher’s LSD multiple comparisons tests as appropriate using GraphPad Prism 7 (La Jolla, California). Data were presented as mean ± SEM, and significance was assumed when * *p* < 0.05. Experimental groups were blinded to the observer, and the code was broken for analysis.

## 3. Results

### 3.1. SynCav1 Preserves Soma Mitochondrial Morphology and Dynamics in Hippocampal Pyramidal Neurons of Symptomatic 12 m PSAPP Mice

Mitochondrial morphology alterations and dysfunction have been well documented in AD [33]. Previous work from our group showed that *SynCav1* significantly delayed memory deficits in PSAPP mice. We, therefore, performed mitochondrial morphometry using EM as previously described. As shown in Figure 2A, 12 m PSAPP-*SynRFP* mice exhibited abnormal mitochondrial morphology consisting of disorganized cristae, matrix swelling and vacuoles, and abnormal distension between inner and outer mitochondrial membranes, morphological indicators of excessive mitochondria damage. Quantification also revealed that 12 m PSAPP-*SynRFP* mice exhibited decreased mitochondria number and increased mitochondrial area. In contrast, 12 m PSAPP-*SynCav1* mice exhibited mitochondrial morphology similar to that exhibited by WT-*SynRFP* mice (Figure 2B,C). Although an increasing trend was observed in mitochondria size in PSAPP-*SynRFP* group at 9 m, no significant difference was found in total mitochondrial size or count among the 3 groups. However, analysis of abnormal mitochondria confirmed a higher percentage of damaged mitochondria in PSAPP-*SynRFP* group at 9 m, with no significant difference between PSAPP-*SynCav1* and WT-*SynRFP* (Figure 2D). Mitophagy dysfunction has been observed in several AD mouse models and is recognized as an important contributor to AD pathogenesis [34]. Quantification of mitophagy-like events in both PSAPP-*SynRFP* and PSAPP-*SynCav1* mice were significantly lower compared to WT-SynRFP mice, with no significant difference between the two PSAPP groups (Appendix A). These findings demonstrate that *SynCav1* preserves hippocampal soma mitochondrial profile (i.e., counts and morphology) in PSAPP mice, independent of the mitophagy-related pathway.

### 3.2. SynCav1 Preserves Synaptic Mitochondrial Number and Morphology in Hippocampal Pyramidal Neurons of 12 m PSAPP Mice

Synaptic mitochondria play a vital role in maintaining the synaptic function, including chemical and electrical transmission. Thus, we further analyzed synaptic mitochondria in the stratum radiatum within the hippocampal CA1 subfield, which includes synapses on apical dendrites of pyramidal cells. Although no significant difference was found in total mitochondrial number at 9 m between groups, quantification revealed a slight increase in mitochondrial area in both PSAPP-*SynRFP* and PSAPP-*SynCav1* mice compared to WT-*SynRFP* mice. At 12 m, synaptic mitochondrial count in PSAPP-*SynRFP* mice was significantly decreased while simultaneously exhibiting larger mitochondrial area with no significant difference between PSAPP-*SynCav1* and WT-*SynRFP*, although mitochondrial area from PSAPP-SynCav1 mice was also significantly larger than mitochondria from WT-*SynRFP* at 12 m. These data suggest that *SynCav1* alleviated synaptic mitochondrial damage in PSAPP mice (Figure 3A–D).

Previous work from our group already demonstrated PSAPP mice showed decreased total synapses and number of synaptic vesicles (PSVs) at 9 m [22]. Because synaptic mitochondria are essential to synaptic maintenance and function, we further analyzed pre-synaptic mitochondria at both 9 and 12 m. Interestingly, no significant difference was found in the ratio of mitochondria-containing pre-synaptic terminals at 9 m among the 3 groups, while 12 m PSAPP-*SynRFP* mice showed a significantly lower proportion of mitochondrial within pre-synaptic terminals compared to WT-*SynRFP* mice (Figure 3E,F), findings consistent with others that demonstrated loss of mitochondria in axons and dendrites in postmortem human AD brains [35]. These changes may explain synaptic dysfunction and neuronal damage in AD. In contrast, PSAPP-*SynCav1* mice showed a higher proportion of mitochondria within pre-synaptic terminals versus PSAPP-*SynRFP*, with no significant difference compared to WT-*SynRFP* mice. These findings indicate that *SynCav1* prevents loss of synaptic mitochondria in 12 m PSAPP mice and may in part explain the cognitive and synaptic preservation previously demonstrated with *SynCav1* gene therapy [22].

### 3.3. Upregulated P-DRP1 in PSAPP Mice Is Inhibited by SynCav1

As revealed by mitochondria morphometric analysis, mitochondria morphology among PSAPP groups was found to vary from small, dense mitochondria to extremely large, swollen mitochondria with disrupted cristae. Fission and fusion events are major regulators of mitochondrial morphology. Thus, we further analyzed the ratio of mitochondria that engaged in either fission or fusion as determined by EM (Figure 4A). At 9 m, although there was an increased trend in fusion/fission events in both PSAPP groups, no significant difference was found among the 3 groups. Consistent with these findings, immunoblot analysis showed no significant difference in the expression of mitochondrial fission and fusion proteins in hippocampal tissue between groups at 9 m (Figure 4B). At 12 m, there was a significant increase of fission/fusion events in PSAPP-*SynRFP* (Figure 3A) alone with elevated phosphorylated (p-)DRP expression in PSAPP-*SynRFP* mice compared to WT-*SynRFP* and PSAPP-*SynCav1* mice (Figure 4C,D), results consistent with that observed in postmortem human brains [36]. Interestingly, PSAPP-*SynCav1* mice also showed a significant increase in mitofusin-1 (Mfn1) expression compared to other groups, which has been observed to be significantly downregulated in brain tissue from AD patients [37]. Such data suggest that neuronal Cav-1 may dynamically regulate mitochondrial fusion/fission, which is critical to increasing neuroplasticity.

### 3.4. SynCav1 Augments Hippocampal Mitochondrial Respiration in Symptomatic PSAPP Mice

Mitochondrial functional plasticity is closely linked to its morphology. Alterations in the fission/fusion machinery, which resulted in abnormal mitochondrial morphology, could significantly affect mitochondrial respiration, which is of particular importance in neurons due to high-energy demands in the brain [24,25,26,27]. Since we observed preservation of mitochondrial morphology and dynamics in PSAPP-*SynCav1* mice, we, therefore, measured mitochondrial function in hippocampal tissue using the Oroboros O2k-Respirometer. In the presence of complex I and II substrates, no significant difference was found in 9 m old PSAPP-*SynRFP* hippocampal homogenates compared to WT-SynRFP (data not shown). A significant decrease in mitochondrial respiration through complex I and II (CI and CII OX), maximum oxidative phosphorylation capacity (CI + II OX), and maximum uncoupled capacity (CI + II mUC) was detected in 12 m PSAPP-*SynRFP* mice compared to WT-*SynRFP* and PSAPP-*SynCav1* mice. No significant difference was found between WT-*SynRFP* and PSAPP-*SynCav1* mice (Figure 5A,B). This proportionally increased oxygen consumption rate through the whole respiratory state confirmed the mass and quality of mitochondria observed in SynCav1. Since we observed mitochondria content changes across groups by EM, we thus further analyzed oxygen flux control rate (FCR), which can avoid the effect from viability in mitochondrial content [32,38]. When regarding the proportion of ETS used via CI and CII separately, a significant reduction was observed in PSAPP-*SynRFP* group compared to PSAPP-*SynRFP* group. On the contrary, PSAPP-*SynCav1* group used a significantly higher part of their maximal ETS capacity via CII and a non-significant increase trend in CI (Appendix A).

To better understand the molecular mechanisms underlying these alterations in mitochondrial respiration, we immunoblotted for proteins in the electron transfer chain (ETC) complex at 12 m. The expression levels of different ETC proteins did not significantly change between the 3 groups, as evident by similar expression levels of oxphos complex proteins (Figure 5C). Further immunoblot analysis revealed no significant difference in Tom 20, a mitochondrial import receptor subunit and marker of mitochondrial mass [39], although a non-significant decreased trend was observed in PSAPP-*SynRFP* group. Such data suggest that in lieu of protein deficits in ETC components, the regulatory function of caveolin in AD pathology may be in maintaining mitochondrial function to allowed for coupled respiration.

## 4. Discussion

The pathophysiology of AD is complex, consisting of amyloid deposits and metabolic dysfunction accompanied by hyper-neuroinflammatory responses, all of which exacerbate synaptic loss and neuronal death. Previously, we demonstrated decreased hippocampal Cav-1 in 9 m PSAPP mice, while hippocampal delivery of *SynCav1* offered neuroprotection independent of reducing toxic amyloid deposits and astrogliosis [22]. The current study more thoroughly elucidated the neuroprotective mechanism of *SynCav1* in PSAPP mice to occur in part through preservation of mitochondria morphology, structural dynamics, and respiratory capacity.

Of all the neuropathological changes observed in AD, the loss of synapse correlates most strongly with cognitive decline. Mitochondria are normally present in areas of high energy demand such as pre-synaptic terminals because synaptic mitochondria provide ATP for synaptic function and mobilization of pre-synaptic vesicles essential for neurotransmission; the spatial distribution of mitochondria in neurons can, therefore, reflect synaptic activity and neuroplasticity. Notably, soma mitochondria still maintain relatively normal morphology and abundance in PSAPP-*SynRFP* at early symptomatic stage (9 m), in contrast to synaptic mitochondria in PSAPP-*SynRFP* mice, which already exhibit a larger area compared to WT-*SynRFP* mice and may explain the early synaptic dysfunction that closely correlated with AD-associated cognitive deficits [40].

The total number and morphology of mitochondria are mainly regulated by the balance of fission and fusion (i.e., mitochondrial dynamics) [41]. While fusion allows for the exchange of lipid membranes and mitochondrial content, fission allows the elimination of irreversibly damaged mitochondria and abnormal content. Mounting evidence reveals that abnormities in mitochondrial dynamics widely precede many of the hallmarks of pathology, which, if appropriately regulated or targeted, could afford significant beneficial effects for patients afflicted with AD. For example, increased p-DRP-1 was observed in several AD mouse models, resulting in excessive fragmentation of mitochondria and subsequent mitochondrial dysfunction and neuronal damage [37]. In the last few years, several DRP1 inhibitors have shown beneficial effects against mutant protein-induced mitochondrial and synaptic damage in neurodegenerative diseases [42,43,44,45]. Here we reveal that *SynCav1* inhibits the significant upregulation of p-DRP1 in PSAPP mice. Furthermore, *SynCav1*-treated PSAPP mice also showed significantly increased Mfn1 expression, another protein that has been targeted for attenuating or reversing axonal degeneration [46]. Mfn1 has previously been shown to be decreased in the same mouse model we used in this study [47]. Furthermore, clinical evidence has revealed that Mfn1 was decreased in AD patients and this decrease correlated with cognitive deficit [37,48]. Although we did not observe decreased Mfn1 in our PSAPP group (a possible reason is the time point of sample collection), the consistent elevated Mfn1 in the *SynCav1* group may, in part, explain our previously published findings that demonstrated preserved axonal structure and cognitive function in 11-month-old PSAPP mice treated with *SynCav1* [22]. These data strongly indicate that the neuroprotective effect from *SynCav1* in PSAPP mice is through maintaining mitochondrial dynamics and morphology. Interestingly, although we observed fewer mitochondria in PSAPP mice by electron microscopy, the gross mitochondrial mass measured by Tom20 did not show any significant difference among the groups, suggesting that the gross indicator lacks the biological sensitivity to detect changes in mitochondrial ultrastructure. Several studies have shown that Cav-1 actively participates in maintaining mitochondrial integrity and function by regulating mitochondrial protein quality control [19,49]. Because mitophagy failure has been considered as a driver of AD pathogenesis [34,49,50,51,52,53], we investigated whether the neuroprotective effect of Cav-1 occurred through mitophagic-mediated elimination of damaged mitochondria. However, mitophagy-like events as measured by EM failed to show any significant difference between the two PSAPP groups, suggesting that *SynCav1* does not affect mitophagy in PSAPP mice (data not shown here).

A growing body of research has highlighted the regulatory role of MLRs and Cav-1 on mitochondrial energetics in non-neuronal systems such as tumors, skeletal, and cardiac systems [5,10,54,55]. However, little research has aimed to understand the mechanistic role of Cav-1 in metabolism in the context of neurodegeneration diseases. Here, we demonstrate a clear correlation between Cav-1 expression and mitochondrial morphology and function in neural systems within AD models. Mounting evidence suggests that Abeta accumulation alone can cause mitochondria dysfunction. The decreased Cav-1 contributes to deficits in mitochondrial quality control, which may exacerbate the ability of neurons to adapt to oxidative stress. Conversely, Cav-1 re-introduction with *SynCav1* significantly rescued mitochondrial dysfunction. The fact that we did not observe much mitochondrial improvement with *SynCav1* in 9-month-old PSAPP mice suggests that neuronal Cav-1 may not affect baseline performance but rather limits the progress of mitochondrial dysfunction at later stages of the disease. These data lend mechanistic insight into the neuroprotective effects of Cav-1 in diseases models of AD. Another potential mechanism through which Cav-1 works may occur through regulating Ca2+ influx, an intracellular event, which has been observed in tumor cells [56], endothelial [57], and smooth muscle cells [58], a process that may couple to mitochondrial resiliency as mitochondria can serve as a calcium sink in cells [59]. Similar effects were observed in a recently published study from Chen and colleagues that showed treadmill exercise protected mitochondrial integrity and inhibited endogenous mitochondrial-mediated apoptosis through Cav-1 upregulation [60]. These results demonstrate a key role in Cav-1 for maintaining mitochondria dynamics and further argue for the overexpression of neuronal Cav-1 as a potential therapeutic strategy to treat neurodegeneration diseases through preservation of mitochondrial function and dynamics.

## 5. Conclusions

In conclusion, this study shows that Cav-1 is a critical regulator of mitochondrial homeostasis in the PSAPP mouse model of AD. Serving as a rapid responder to oxidative stress, Cav-1 has been proven to be essential for proper quality control for maintaining mitochondria homeostasis. Decreased Cav-1 expression in neural systems in PSAPP mice leads to failure of mitochondrial quality control and accumulation of damaged mitochondria, specifically in synaptic regions. Loss of synaptic mitochondrial function invokes severe disturbances in synaptic function, which is preserved with *SynCav1* gene transfer. Whether *SynCav1* can be exploited as a therapeutic target to treat humans afflicted with AD needs further investigation.

## 6. Patents

Synapsin-caveolin gene therapy is patented by the U.S. Patent office (US 8,969,077 B2) and owned by the Department of Veterans Affairs (10-084) and UCSD (2010-117-0).

## Figures and Tables

**Figure 1 cells-10-02487-f001:**
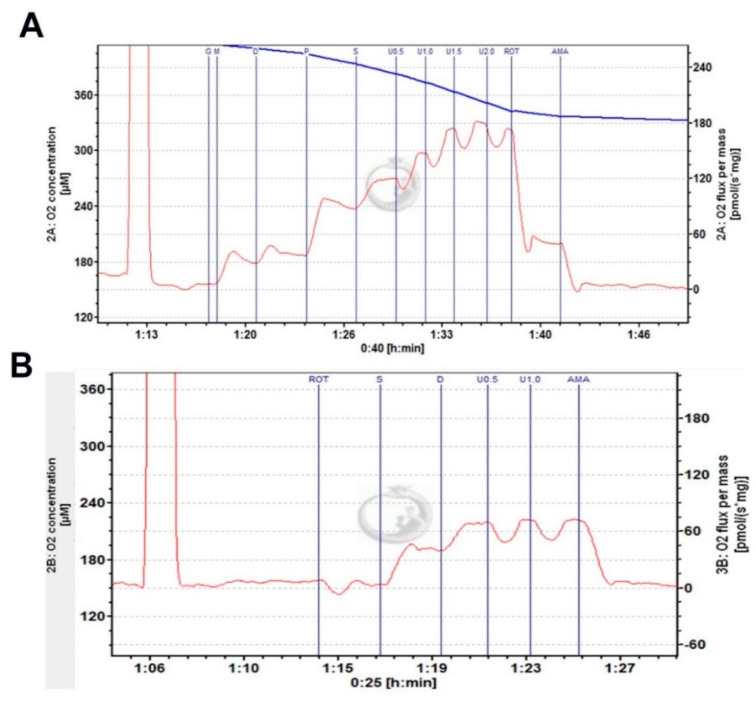
Representative respirometry traces of hippocampal homogenates using the Oroboros O2k-Respirometer. (**A**) Complex I and II linked activity. (**B**) Complex II linked activity. Complexes substrates as described in the methods.

**Figure 2 cells-10-02487-f002:**
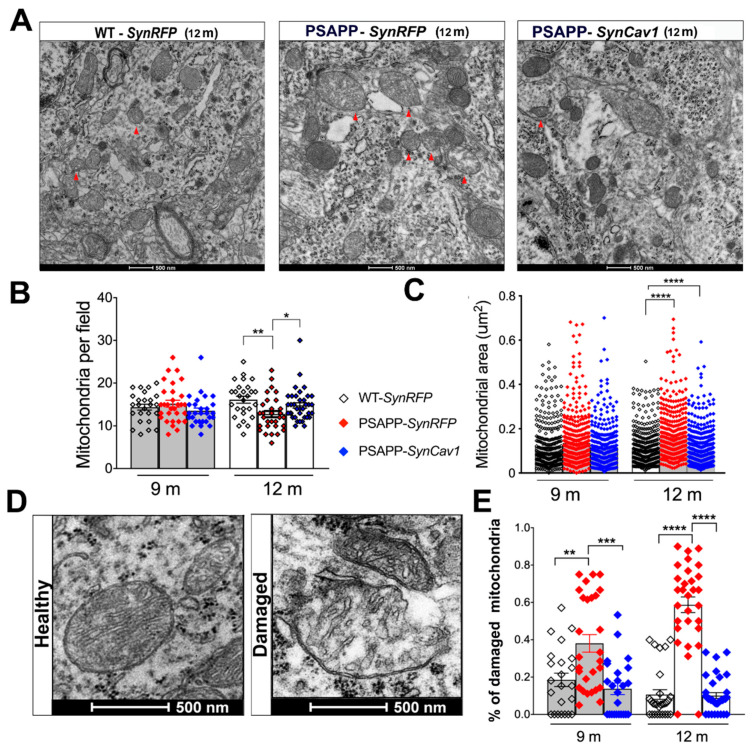
PSAPP-*SynRFP* mice exhibit less soma mitochondria with larger area, while PSAPP-*SynCav1* shows a similar mitochondria profile as WT-*SynRFP*. (**A**) EM images of CA1 molecular cell layer consisting of pyramidal cell bodies in 12 m mice. The red arrow indicates damaged mitochondria with disrupted cristae and empty vacuoles. Quantitation of mitochondrial count and area (**B**,**C**). Representative EM image and quantification of damaged mitochondria (**D**,**E**). Data are presented as mean ± S.E.M., *n* = 5 animals per group with 8–12 micrographs per animal. Data were analyzed using One-Way ANOVA. Significance was assumed when *p* < 0.05. * *p* < 0.05, ** *p* < 0.01, *** *p* < 0.001, **** *p* < 0.0001.

**Figure 3 cells-10-02487-f003:**
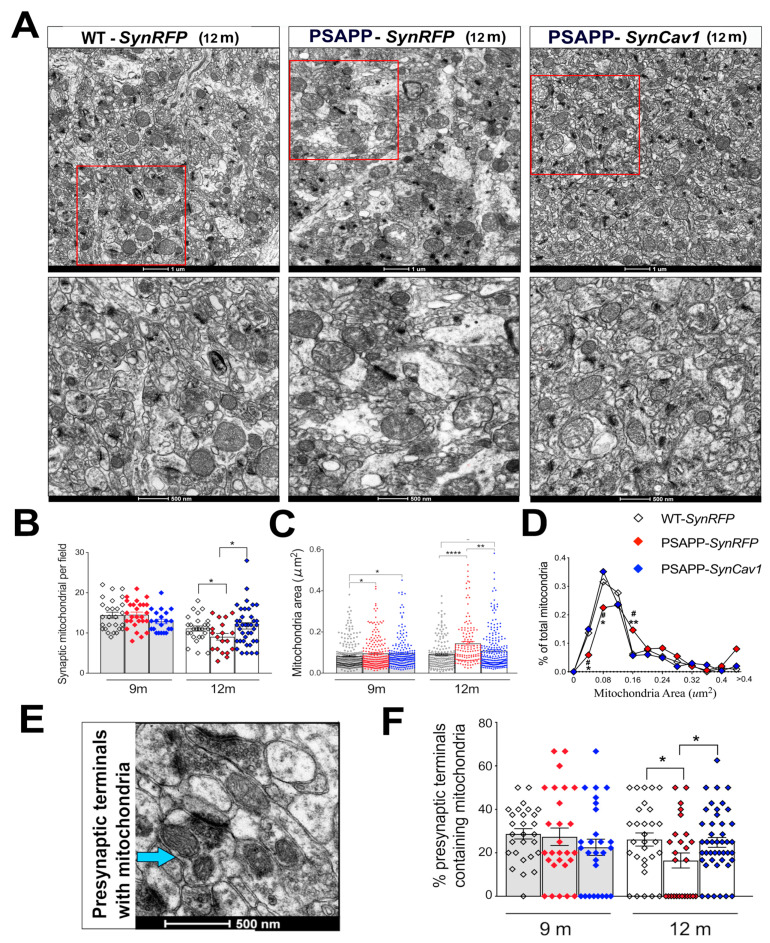
*SynCav1* preserves synaptic mitochondrial number and morphology in PSAPP mice**.** (**A**) Representative images of synaptic mitochondria in CA1 distal apical dendrites within the stratum radiatum with high magnification of representative region at 12 m. Quantitation of synaptic mitochondrial number and area (**B**,**C**) and % of total mitochondria relative to the area (**D**). Representative image and quantification (**E**,**F**) of mitochondria-containing pre-synaptic terminal. Data are presented as mean ± S.E.M., *n* = 5–7 animals per group with 10 micrographs per animal. Data were analyzed using One-Way ANOVA. Significance was assumed when *p* < 0.05. * *p* < 0.05, ** *p* < 0.01, **** *p* < 0.0001.

**Figure 4 cells-10-02487-f004:**
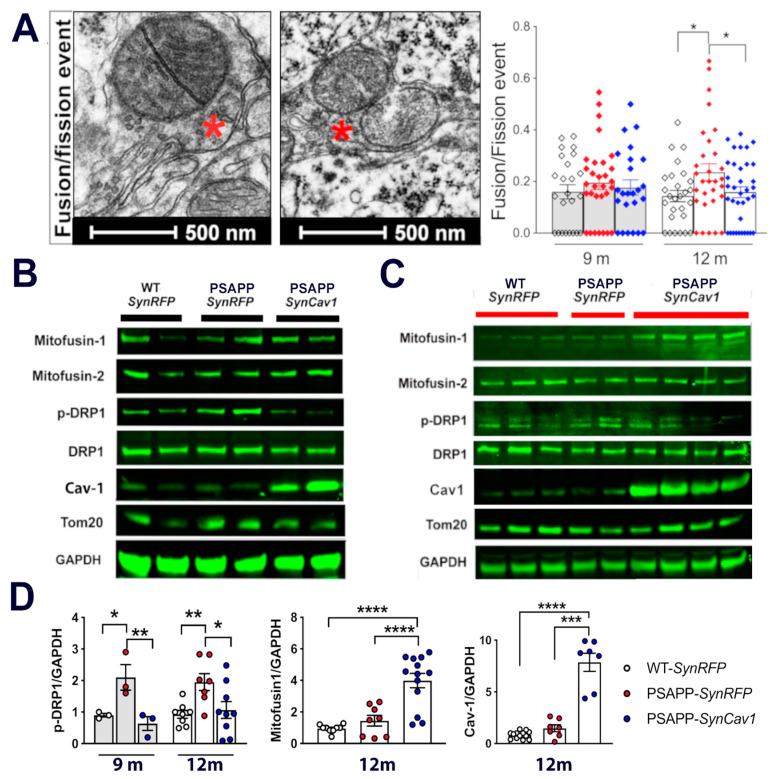
*SynCav1* inhibits excessive mitochondria fission observed in 12 m PSAPP mice hippocampus. (**A**) Represent image from 9 m. (**B**) and 12 m (**C**). (**D**) Quantification of immunoblots at 12 m. Data are presented as mean ± S.E.M., *n* = 3 per group for 9 m, *n* = 6–10 animals per group for 12 m. Data were analyzed using one-way analysis of variance (ANOVA) followed by Fisher’s LSD multiple comparisons test. Significance was assumed when *p* < 0.05. * *p* < 0.05, ** *p* < 0.01, *** *p* < 0.001, **** *p* < 0.0001.

**Figure 5 cells-10-02487-f005:**
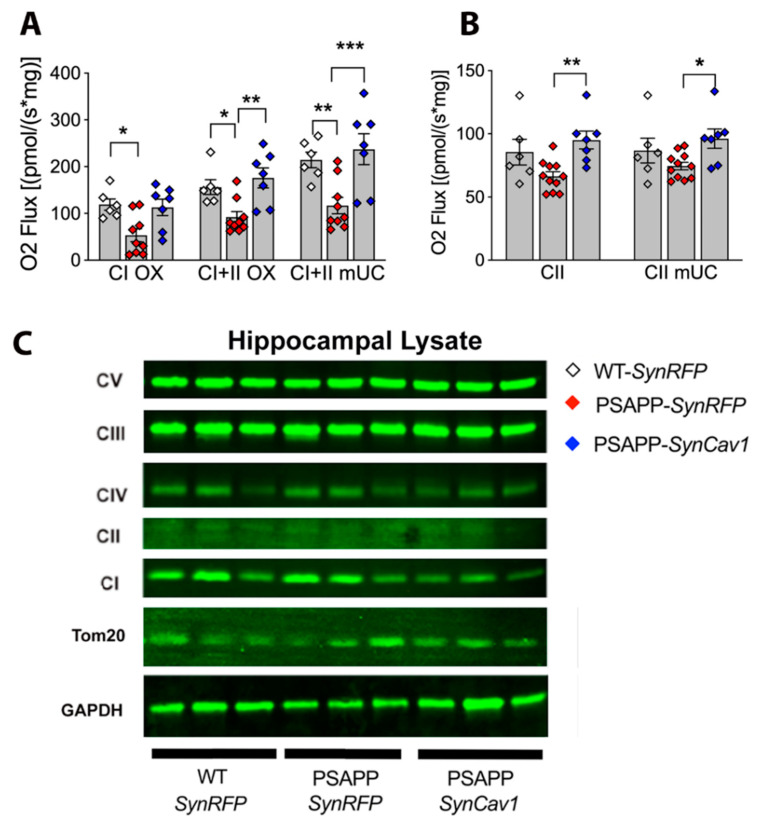
*SynCav1* preserves hippocampal mitochondria respiration through complex I and II, maximal oxidative phosphorylation and maximum uncoupled capacity in 12 m PSAPP mice. (**A**,**B**) Quantification of tissue-specific oxygen flux. (**C**) Presentative blots at 12 m. Data are presented as mean ± S.E.M., *n* = 6–10 per group. Data were analyzed using one-way analysis of variance (ANOVA) followed by Tukey multiple comparisons test. Significance was assumed when *p* < 0.05. * *p* < 0.05, ** *p* < 0.01, *** *p* < 0.001.

## Data Availability

Data available upon request from the corresponding authors. Data are not publicly available due to research policies guided by the Department of Veterans Affairs.

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
