# Peer review of "Synapsin-Promoted Caveolin-1 Overexpression Maintains Mitochondrial Morphology and Function in PSAPP Alzheimer’s Disease Mice"

_cells, 2021, doi:10.3390/cells10092487_

Round 1
Reviewer 1 Report
In this current manuscript Wang and al, report the impact of Caveolin-1 expression on mitochondrial morphology and function in PSAPP Alzheimer’s Disease mice.
Although, molecular mechanisms involved in mitochondrial dysfunctions have a strong interest for AD process, this manuscript seems to be more a follow-up of the previous publication [20]. Indeed, the link between cav1 expression and mitochondrial phenotypes linked to AD seems to be based on unexpected observations than investigations following a clear hypothesis. Overall, the results are clear, well presented and supported by data generated using relevant analysis. However, the authors should clarify several points to increase understanding of their intellectual approach.
First of all, the authors did not show any data about the morphology and function of mitochondria in Syn-Cav1 mice that do not express PSAPP. Indeed, it should be helpful to see if the Cav1 expression alone does not modify these parameters.
Moreover, within discussion, the authors should discuss why the impact of PSAPP on mitochondrial dysfunction could be dependent of Cav1-expression.
Second, it is not clear how the authors have chosen to investigate protein expressions. Because proteomic analysis was performed on syn-Cav1 mice [20], did the authors crossed check with these differently-expressed proteins and mitochondrial functions? It should be interest to bring this information to support the importance of Cav1 expression in mitochondrial functions in the context of PSAPP expression.
Manor point:
Sentence 30- The authors claim a link between neurodegenerative disease, loss of neuroplasticity and down-regulation of Cav-1 expression without reference. References [19-20] are associated with from mice models and should not be used to support this affirmation. The authors should use references based on human data or clarify the hypothesis.
Sentence 38- Reference should be added to support “the contribution of oxidative stress to ALS and AD neuropathology”.
Reviewer 2 Report
Wang and colleagues have characterized mitochondrial morphology/functionality in a mouse model of Alzheimer’s disease by restoring Cav-1 expression under synapsin promoter control. Using mainly electron microscopy, authors show that in the presence of SynCav1, mitochondria are less disorganized and most numerous, especially at the synaptic level. Furthermore, the mitochondrial respiration is enhanced.
The main issue with this manuscript is that it appears as a preliminary and incomplete work of characterization in a model familiar to the authors. They merely described what they observed, without investigating the mechanism. Also, omitting several methodological issues described below, the respirometric data are just confirmative. In the presence of well-organized and most abundant mitochondria, a general enhancement of all the respiratory states is a largely expected consequence. Finally, several data appears conflicting or not properly extensively analyzed.
Following my specific comments:
1) Authors made large use of electron microscopy, a technique perfectly is suitable for morphological and quantitative analysis. How they used it for determining mitophagy? Also, what the authors are looking at in the Supplementary Figure? They just talk about generic “mitophagy-like events” without no explanation neither in the Method nor in the Results section. This makes this part difficult to comprehend. As they did later for the fusion/fission events, authors should confirm electron microscopy experiments with the analysis of mitophagy markers (PINK/Parkin, LC3).
2) Data concerning fission/fusion indicate that SynCav1 reduces the level of p-DRP1 in AD mice up to similar value to WT ones. This modulation could correlate with some positive effect, as suggested by the literature. On the other hand, SynCav1 promotes a significant increase of Mitofusin-1 in AD mice at both 9 and 12 mo. Why this occurred? How it can be interpreted, it is positive or negative for the cells? To me, it seems to create some imbalance in the two process that authors did not properly deepened or discussed.
3) The SUIT protocol used by the authors was not properly described. Also, the reference cited by them (n. 22) doesn’t even do it. Overall, paragraph 2.5 is confusingly written and it doesn’t help in the understanding of the experiments’ rationale. It is extremely important to provide a detailed description of the protocol, with the precise order of substrates addition in the cuvette, their concentration, etc. Showing a representative curve with all the addition could be helpful.
4) Respirometric experiments presents some methodological issues. Why the pyruvate was not used among the NADH-linked substrates? It is well known that pyruvate, malate and glutamate are all able to stimulate complex I. In which conditions complex II was evaluated? As recently demonstrated, the oxaloacetate concentration directly linked to complex I can significantly modulate complex II activity (see doi: 10.3390/ijms21217809).
5) Overall, the respirometric results are confirmative of the increased amount in term of mass and quality of mitochondria observed in SynCav1. This is suggested by the fact that oxygen consumption grows proportionally in all the respiratory state. In this sense, the data is not very informative. I would suggest to analyze the flux control ratio rather than the oxygen consumption (see doi: 10.1016/j.biocel.2009.03.013). This sometimes is much more informative and allow to understand the contribution of CI, CII or a specific respiratory state to the maximal capacity.
6) In my opinion, blots in Fig. 4C appear a big issue since they are in conflict with the previous electron microscopy data and with the assumption of the whole manuscript. If an increased in mitochondrial amount was showed for SynCav1, a lower Tom20/GAPDH ratio in SynRFP mice was expected.
Based on the above, I believe that, at this stage, the manuscript is not ready for publication in this journal.
Reviewer 3 Report
Minor comments
- The order of the Results section is incorrect: 3.1, 3.2, 3.5, 3.4, and paragraph 3.3 is missing.
- References are need at line 187 –190, 211–213, 219-220.
- A graphic abstract is required for easy understanding of this paper.
Major comments
- Please check the effect of overexpressing Cav-1 on mitochondria in the control group.
- In a previous study, Cav-1 was decreased in 9-month-old PSAPP mice, but in this study, Cav-1 was higher in 12-month-old PSAPP mice than in WT mice (fig. 3D). The authors need further explanation of this result.
- The authors described that mitochondrial dysfunction plays a major role in AD. However, detailed descriptions of mitochondrial dysfunction in AD are lacking. Therefore, it seems that an additional description of mitochondrial dysfunction in AD is necessary in introduction.
- The authors need additional experiments to demonstrate the neuroprotective effect induced by SynCav1 using examination of neuronal cell death.
- The authors conducted stereotaxic injection of red fluorescent protein (RFP) into PSAPP mice using AAV9 virus in this study. So, data showing whether the virus-injected site was targeted region are required using RFP fluorescence.
- The authors overexpress Cav-1 using a virus in PSAPP mice and show its expression by WB: Fig. 3B-D. However, to clearly show that the expression of Cav-1 increased in a region (hippocampus)-specific manner, histological staining is appropriate rather than WB.
- In this study, there was no mitochondrial improvement effect due to Cav-1 overexpression in 9-month-old PSAPP mice. Please describe this results in the discussion section more sufficiently.
Reviewer 4 Report
Wang S. and colleagues showed that Cav-1 is a critical regulator of mitochondrial homeostasis in the PSAPP mouse model of AD. Decreased Cav-1 expression in neural systems in PSAPP mice leads to failure of mitochondrial quality control and accumulation of damaged mitochondria, specifically in synaptic regions. Loss of synaptic mitochondrial function invokes severe disturbances in synaptic function which is preserved with SynCav1 gene transfer. This study focuses on one of early events in neurodegeneration the mitochondria homeostasis. Data are interesting and the structure is good, but it needs improvements in the form (labeling and font) and in the experimental plan/analysis.
Minor Comments:
- Title: Synapsin-caveolin in the title is misleading the reader. It will lead to a manuscript where the interaction/pathway between two proteins: Synapsin and Caveolin maintains mitochondrial morphology and function. SynCav1 system used here is the system to overexpress the Caveolin1.
- Clarify in the abstract what is the function of p-DPR1 and Mfn1
- Line 45-46 maybe re-phrase as: “…can cause synaptic dysfunction and cognitive decline, pathological changes observed in AD”
- Include in the introduction PSAPP pathology description as amyloid and tau deposition timeline, whit this also specify why 9- and 12-months old animals have been used for the EM and analysis?
- Figure 2. it is missing E in the Figure 2; it should be on the quantification.
- Title 3.3 looks contradictory “Elevated….is reduced…”. It is elevated compared to…? I suggest eliminating
- Discussion: how Caveolin-1 (cell membrane associated protein) modulates mitochondria? Is it located also in the mitochondria? If not which theory the authors have on a possible transduction of signals.
Major Comments:
- It is important describe in a separate section (within Materials and methods) the analysis and quantification of mitochondria from EM. Statement as “Alterations in mitochondrial morpho-106 logic profile were measured using Adobe Photoshop (San Jose, CA, USA) as previously 107-108 Line” is not precise. No reference is reported and no details on analysis are reported. Which pipeline has been used to analyzed alterations? Usually, ImageJ FIJI or Imaris Software are used to design one. The author mentioned alterations, please specify which alteration type (breaks, invagination etc…). it’s important be specific.
- Figure 2D. Describe which type of mitochondrial damage has been analyzed and quantified. As the author described several alterations are observed.
- Which is the reason beacuse the injection of SynRFP (so expressing only RFP) cause so much damage in the mitochondria? In the quantification are missing un-injected group so it becomes complicate interpret the data.
- Supplemental Figure 2 need improvements in the text and in the labeling. Line 168 Suppl Figure 2E is mentioned but not present in the label E in the supplemental file.
- Specify why CA1 synapses have been analyzed and not other brain regions. Is it particularly affected in this model?
- Lane 207-210. Authors claim morphometric analysis including dense mitochondria, mitochondria with disrupted cristae that are not present in the present study as analysis as well as images. I suggest adding a figure where all the alterations observed are represented with arrows and highlighting lines to show morphological changes.
Round 2
Reviewer 2 Report
Although I appreciate the efforts made by the authors, my opinion is not changed after the revisions. As I previously wrote, this work appears too preliminary for the publication in a journal like this. Furthermore, authors did not addressed properly the previous comments. For example, they now have described the oxygraphy protocol but still the data are just confermative and very low informative, and FCRs (that could probably give them new information) were not
Reviewer 3 Report
- The mice used in the previous study were 11 months old, and the mice used in the present study were 12 months old. Therefore, in the present study, authors should be conduct additional experiment in 12-month-old wild-type mice.
- Authors examined the neuroprotective effect of SynCav1 in AD. However, neuronal cell death in the ischemia and trauma model is different from neuronal cell death in AD. So, authors need additional experiments to demonstrate the neuroprotective effect induced by SynCav1 using examination of neuronal cell death in AD.
- The virus-injected site we requested is not the previous study, but the needle track administered in present study.
- The authors described in introduction part that mitochondrial dysfunction plays a major role in AD. However, the content is not sufficient for explain the relation of AD and mitochondria. Please describe in more detail of mitochondria dysfunction occurs in AD.
- Lines 184-185: Additional references or experiments are needed to support that SynCav1 gene therapy prevents cognition and synaptic preservation.
- line 298: "These data lend mechanistic insight into the neuroprotective effects of Cav-1 in diseases models of AD and ALS."
In this study, authors examined the neuroprotective effect of SynCav1 in animal model of AD, not ALS. So, it is out of context that Cav-1 exhibits neuroprotective effects in ALS.
Round 3
Reviewer 2 Report
Again, I appreciate the authors' effort to improve the manuscript. However, as stated in previous revision rounds, the main problem with the manuscript is that it appears as a preliminary work. In my opinion, it is not suitable for publication in a journal with a high impact factor at this stage.
Author Response
None
Reviewer 3 Report
Accept in present form
Author Response
None